# Elastomeric Polyurethane Foams Incorporated with Nanosized Hydroxyapatite Fillers for Plastic Reconstruction

**DOI:** 10.3390/nano8120972

**Published:** 2018-11-25

**Authors:** Lili Lin, Jingqi Ma, Quanjing Mei, Bin Cai, Jie Chen, Yi Zuo, Qin Zou, Jidong Li, Yubao Li

**Affiliations:** Research Center for Nano-Biomaterials, Analytical and Testing Center, Sichuan University, Chengdu 610064, China; linll900730@163.com (L.L.); jingqima2018@gmail.com (J.M.); scumqj@foxmail.com (Q.M.); haisengyan@sina.com (B.C.); ChenJie005@outlook.com (J.C.); zouqin80913@126.com (Q.Z.); nic1979@scu.edu.cn (J.L.)

**Keywords:** elastomeric foam, plastic reconstruction, hydroxyapatite/polyurethane, nanosized dispersion, viscoelasticity, biocompatibility

## Abstract

Plastic surgeons have long searched for the ideal materials to use in craniomaxillofacial reconstruction. The aim of this study was to obtain a novel porous elastomer based on designed aliphatic polyurethane (PU) and nanosized hydroxyapatite (n-HA) fillers for plastic reconstruction. The physicochemical properties of the prepared composite elastomer were characterized by infrared spectroscopy (IR), X-ray diffraction (XRD), scanning electron microscopy with energy dispersive X-ray spectroscopy (SEM-EDX), transmission electron microscopy (TEM), thermal analysis, mechanical tests, and X-ray photoelectron spectroscopy (XPS). The results assessed by the dynamic mechanical analysis (DMA) demonstrated that the n-HA/PU compounded foams had a good elasticity, flexibility, and supporting strength. The homogenous dispersion of the n-HA fillers could be observed throughout the cross-linked PU matrix. The porous elastomer also showed a uniform pore structure and a resilience to hold against general press and tensile stress. In addition, the elastomeric foams showed no evidence of cytotoxicity and exhibited the ability to enhance cell proliferation and attachment when evaluated using rat-bone-marrow-derived mesenchymal stem cells (BMSCs). The animal experiments indicated that the porous elastomers could form a good integration with bone tissue. The presence of n-HA fillers promoted cell infiltration and tissue regeneration. The elastomeric and bioactive n-HA/PU composite foam could be a good candidate for future plastic reconstruction.

## 1. Introduction

Plastic materials have a huge clinical market for patients of plastic, reconstructive, and aesthetic surgery [1]. The development of orthopedic materials plays a crucial role in plastic surgery, which benefits from the spring-up of new materials [2]. Maxillofacial, oral, and plastic surgeons keep on looking for ideal materials, which should have osteointegration potential, biological safety, easy molding and good plasticity, appropriate mechanical support, and should be cost-effective for skeleton reconstruction and shape restoration [3,4]. In the late of 1960s, W. L. Gore invented expanded polytetrafluoroethylene (e-PTFE), which has become a popular material in the field of facial plastic and reconstructive surgery [5]. The e-PTFE sheets have been used in the temporal region and have achieved good results, of which, the physic trait of the e-PTFE sheet is similar to human soft tissue [6]. Moreover, hyaluronic acid is another useful material for tissue augmentation, because of its unique and favorable biophysical properties [7]. However, the use of hyaluronic acid is very limited because of its very short lifetime, and because of its rapid elimination from the injection site in less than a week [8]. It is regretful that both the e-PTFE and the hyaluronic acid are not bioactive and cannot form a bond with bone tissue that easily introduces the implant migration after implantation.

Calcium phosphate mimics the osteoid compound of mature bone, which is a foundation of the bioactive techniques for alloplastic cranioplasty [9]. The excellent biocompatibility and bone-bonding bioactivity of synthetic hydroxyapatite (HA) have made it attractive in plastic surgery. Yet, the pristine ceramic based on inorganic HA is too brittle, and it is difficult to mold or tailor in the operation [10]. In recent years, hybrid composites composed of polymers with nanosized hydroxyapatite (n-HA) may create a bio-mimetic interface to host tissue, and can be considered as one of the most promising biomaterials. Different n-HA/polymer composites, such as n-HA added poly(lactic-co-glycolic acid) (PLGA) or polycaprolactone (PCL), have been developed and broadly investigated as scaffolds for bone repair [11,12,13]. But few of them have been used for plastic and reconstructive surgery, because the specific biomaterials need match to the comfortable and aesthetic requirements of tissue augmentation, such as resilience, adjusting physical softness and hardness, contact pressure, uniform block, and easy shaping [14,15]. It is well known that there is a class of polymer materials that can be synthesized as “soft or hard scaffolds” by different segmented structures [16]. The synthetic polymer called polyurethane (PU) has been applied in tissue engineering with the advent of a wide range of mechanical and physical properties by changing the ratio of soft and hard segments and the composition [17,18]. PU exhibit diverse properties with a self-foaming porous and unique segmented structure, excellent and varying mechanical properties, and good biocompatibility, which could be considerable interest to specific tissue [19,20]. In our previous studies, several kinds of polyurethane (PU) rigid foams compounded with n-HA nanoparticles have been developed, chiefly for bone repair and regeneration. As a result, the crosslinked structure based on castor oil as a soft segment, has displayed a high modulus for the PU scaffolds for bone regeneration [21,22]. But for plastic surgery, the moderate flexibility of the PU scaffolds is the potential and preferable characteristic for biomedical applications.

In this study, we designed and prepared a foam based on the elastomeric PU matrix and n-HA crystals, to match with the multifunctional requirements for plastic reconstruction. The elastic foam was fabricated in a non-toxic manner, and the obtained product was expected to be easily tailored and to exhibit both enough mechanical support and elastic adaptability, as well as to provide a bioactive microenvironment for cell growth and tissue regeneration. The elasticity of the scaffolds, including resilience and stretchability, has been deeply studied, for the deformation ability is one of the major factors to consider for plastic surgeons. The elastic n-HA/PU foams have been postulated to have a promising prospect for future plastic surgery or reconstruction fields of specific types of tissue.

## 2. Materials and Methods

### 2.1. Materials

Raw reagents that polytetramethylene ether glycol (PTMEG) as a soft segment, isophorone diisocyanate (IPDI) as hard segment, and 1,4-butanediol (BDO) and dibutyltin dilaurate (DBTL) as a chain extender and catalyst, respectively, were purchased from Aladdin Industrial Corporation (Shanghai, China). The n-HA particles were prepared through a chemical precipitation method, as reported in our previous study [23]. All of the reagents were of analytical reagent grade.

### 2.2. Preparation of Porous Elastomer Foams

The following three types of polymeric foams were prepared: (1) pure PU, (2) PU with 10 wt% n-HA (10HA/PU), and (3) PU with 20 wt% n-HA (20HA/PU). Considering that a higher content of inorganic HA will produce a stiffer scaffold, which is not proper for the plastic elastomer, the maximum content of HA has been set to 20 wt%. The porous elastomers were prepared in a three-necked flask under a nitrogen atmosphere. Briefly, PTMEG and IPDI were first mixed thoroughly at 70 °C. The stoichiometric ratio of [NCO]/[OH], called the isocyanate index, was set at 1.5. Then, certain amounts of BDO, DBTL, and distilled water were added successively as chain extender, catalyst, and foaming agent, respectively [24], followed by polymerization at 70 °C for 2 h under stirring. For the composite foam, the n-HA powder was filled in the mixture during the polymerization process. After polymerization, the mixture was finally formed and dried in an oven at 90 °C for 24 h, in order to obtain the porous PU scaffold or the n-HA/PU scaffolds. The scaffolds were cut into discs (φ 14 mm × 1.5 mm) for in vitro cell culture and (φ 6 mm × 1.5 mm) for in vivo studies.

### 2.3. Characterization Analyses

Scanning electron microscope (SEM, JEOL 6500LV, JEOL, Tokyo, Japan) with an energy dispersive spectrometer (EDS, INCA ENERGY 250, Oxford, UK) was used to observe the structure and the Ca and P element mapping of the samples. The test samples were lamellar squares (10 mm × 10 mm × 1 mm) and were sputter-coated with gold before the examination. Transmission electron microscopy (TEM, Tecnai, G2 F20, Hillsboro, OR, USA) was used to observe the morphology of the n-HA crystals. To clearly observe the hard and soft segmented structure of the PU matrix, cryoultramicrotomy with a diamond knife was used to obtain a 50 nm thickness ultrathin section. The microphase image of the PU scaffolds was labeled using a staining agent of ruthenium tetroxide (RuO_4_), for which the staining and exposure time was 15 min [25].

The chemical groups of the porous materials were recorded using Fourier transform infrared spectroscopy (FTIR, Nicolet 6700, Nicolet Perkin Elmer Co., Waltham, MA, USA), at a wavenumber range of 650–4000 cm^−1^. The crystallographic structures of the PU composites were tested using X-ray diffraction (XRD) (PANalytical B.V, EMPYREAN, Almelo, Netherlands) with a Cu Kα radiation. The conditions were 40 kV and 25 mA, and the 2θ scan range was from 10° to 60° at a step size of 0.03°. The element spectra of the PU scaffolds were tested using X-ray photoelectron spectroscopy (XPS, AXIS Ultra DLD, Manchester, UK.) at a step of 1000 m eV with a mono Al X-ray source (150 W). All of the above test samples were lamelliform at a size of 10 mm × 10 mm × 1 mm. The thermal stability of the PU composites was measured using a thermogravimetric analysis (TGA) simultaneous thermal analyzer (STA 449 F3 Jupiter^®^, NETZSCH, Bavaria, Germany), under nitrogen flow rate at 20 mL/min. The TGA temperature range was set from 50 to 600 °C at 10 C/min, after all of the PU scaffolds were dried at 60 °C for 48 h in a vacuum oven so as to eliminate the humidity, and then samples of about 4–5 mg were placed in the aluminum crucible for testing.

The three-dimensional (3D) structure of the scaffold block (φ 6 mm × 12 mm) was reconstructed using a vivaCT80 micro-CT imaging system (SCANCO Medical AG, Brüttisellen, Switzerland). Three groups of foams were scanned, of which a slice increment of 20 μm and a total of 600 microtomographic slices were taken for each sample. The scanning system was set at 45 kV and 175 μA. In addition to the visual assessment of the structural images, the morphometric parameters were directly measured using 3D morphometry from the micro-tomographic datasets, including the density and the fraction of the bone volume/total volume (BV/TV), which means that the materials volume/total volume, and the porosity of the elastomers could be calculated according to the following formula:Porosity_CT_ = (1 − (BV/TV)) × 100%(1)

The opening pore porosity of the foams was also tested using a liquid displacement method. The scaffold samples had a volume (V) of 10 mm × 10 mm × 20 mm, and their weight (M) was recorded. After the samples were immersed in distilled water, a series of evacuation–repressurization cycles were conducted in order to exhaust the gas and force the liquid into the pores of the scaffolds, until the water was squeezed into all of the opening pores of the porous foams. Then, the weight of the scaffolds was recorded as M_0_. The porosity was calculated according to the following formula:Porosity = (M_0_ − M)/V × 100%(2)

The apparent density (ρ) was determined from the mass and volume data of the samples, and was calculated by the following equation:ρ = M/V(3)

Then, the weight and volume of the scaffolds were recorded as M and V, respectively. Five samples were tested in each group.

To study the resilience of the PU foams, the samples were characterized using dynamic mechanical thermal analysis (DMTA) on DMA Q800 (NETZSCH, Bavaria, Germany). The specimens with a dimension of 40 mm × 10 mm × 1 mm were uniaxially deformed in tension mode at a 1 Hz oscillating frequency. The temperature range was −50 to 130 °C at a rate of 3 °C/min. Three samples were tested in each group.

To study the stretchability of the PU elastomers, the tensile strength and Young’s modulus of the elastomeric PU scaffolds were tested using a universal testing machine (AG-IC 50KN, SHIMADZU, Kyoto, Japan). The standard of ASTM D695-96 was adopted, and the dimension of the samples was 40 mm × 10 mm × 1 mm. All of the tests were conducted at the speed of 15 mm min^−1^, with a load capacity of 250 N at room temperature and the typical humidity of the laboratory air was about 80%. Five independent samples were tested for each group.

### 2.4. Cell Culture

Cell culture and proliferation: Bone marrow mesenchymal stem cells (BMSCs) isolated from Sprague Dawley rats (one-month-old, male), with a weight of approximately 100 g, were provided by the West China Animal Center of Sichuan University, and the third passage of the BMSCs were utilized in the experiments [26,27]. After ultrasonic rinsing (SB3200D, Ultrasonic cleanser, Ningbo Xinyi, Ningbo, China) in distilled water and autoclave sterilization (GI54DWS, Autoclave, Schneider Electric, Rueil, France), the PU porous specimens were co-cultured with BMSCs (2 × 10^4^ cells/well) in 24-well plates equilibrated in α type minimum Eagle’s medium (α-MEM) medium (Gibco, 1 mL/well) in a humidified incubator (37 °C, 5% CO_2_). The tissue culture plastic was set as the blank control group for comparison. After seeding, the media was replaced every 48 h for the next 20 days. After 1, 4, 7, and 14 days, the cell proliferation was evaluated using a Cell Counting Kit-8 (CCK-8) kit (Sigma-Aldrich Co., St. Louis, MO, USA). The absorbance values of the water-soluble tetrazolium salt were detected at 450 nm using a multi-label counter (Wallac Victor3 1420, PerkinElmer Co., Waltham, MA, USA). Five samples were tested in each group.

Cell morphology: The extracted solution of the PU and HA/PU samples was used to culture the BMSCs for cell morphology observation, according to the leachate method specified in GB/T 16886.5-2003. The morphology and spreading of the BMSCs growing in the extracted solution of the samples were imaged using a laser scanning confocal microscopy (LSCM; Nikon, A1R, MP+, Toyko, Japan). The BMSCs cultured with the extracted solution at 37 °C for 96 h in a humidified atmosphere of 5% CO2 and 95% air, in between, changed the extracted solution in two days. After incubation, the cells were fixed in a 3.7% (*w*/*v*) formaldehyde solution; permeabilized with 0.2% (*v*/*v*) Triton X-100/phosphate buffered saline (PBS); and stained successively with 0.25 μM MitoTracker^®^ Green FM probes for chondriosome, 0.15 μM Alexa Fluor^®^ 532 phalloidin for red F-actin, and 5 μg/mL blue fluorescent Hoechst 33342 in 1% (*w*/*v*) bovine serum albumin (BSA)/PBS for the nucleus, according to the manufacturer’s instructions of Thermo Fisher Scientific Co (Waltham, MA, USA). The cells were then washed with PBS to remove the excess stains. The labeled cells were imaged using LSCM.

Cell differentiation: A quantitative analysis of the ALP and OCN activity for the BMSCs cells cultured on the three scaffolds using enzyme-linked immunosorbent assay (ELISA) has been employed to assess the osteogenic differentiation study [21]. Each sample was seeded with 1 mL of the cell suspension (2 × 10^4^ cells/per scaffold) in 24-well plate on the first day, then the media was replaced every 48 h over the next 20 days. The culture media contained an osteogenic medium (50 μg/mL L-ascorbic acid, 10 nM dexamethasone, and 10 mM β-glycerophosphate) used to promote the osteoblastic differentiation of the BMSCs. The activity of the alkaline phosphatase (ALP) and the content of osteocalcin (OCN) in the BMSCs were analyzed on day 7, 14, and 21. The samples were firstly washed with PBS three times after the culture media were removed, and afterward, the cells were lysed with 0.2% *(v/v*) Triton X-100/phosphate buffered saline (PBS) on the sample, overnight at 4 °C. The ALP and OCN were quantified separately using ALP activity kits and OCN content assay kits (Shanghai MLBIO Biotechnology CO., Shanghai, China). A series of dilutions were prepared in order to obtain the standard curve, and the absorbance at 450 nm was read using a microplate reader [28,29,30].

### 2.5. Osteogenic Capacity 

Implantation in mandible defects: Twelve rabbits (New Zealand white, male or female was random, about 2.5 kg) were employed in the animal experiments. The implantation periods were set for 4 and 12 weeks. At each time point, two rabbits were in each group, and three groups, were operated on, according to the control, PU, and 20HA/PU groups. The animal experiments on the PU scaffolds were approved by the Animal Ethics Committee of Sichuan University, according to all of the regulations. After the PU and 20HA/PU foams were cut into discs with a diameter of 6 mm and a height of 1.5 mm, the samples were sterilized by autoclave 15 min, three times before use. A defect similar in size to the sample (φ 6 mm × 1.5 mm) was created on both sides of the mandible of each rabbit, and the foam sample was implanted into the defect of each experimental group. Then, the muscle and skin incisions were sutured layer-by-layer. An intramuscular injection of 1 × 10^5^ units of penicillin was given to each rabbit during the initial three days after the implant operation. At the time points of 4 and 12 weeks, the rabbits were sacrificed via CO_2_ asphyxiation, and the samples with the surrounding tissue were harvested.

Histological observation: The collected implantation samples were fixed with 4% buffered paraformaldehyde, decalcified, dehydrated by gradient ethanol, cleaned in xylene, and embedded with paraffin wax. The samples were stained with a hematoxylin-eosin (HE) staining agent (Thermo Fisher, Waltham, MA, USA) after being cut into thin sections (5 μm) along the sagittal plane, and were observed under optical microscopy. A schematic diagram showing the material preparation and animal experiment is given in Scheme 1.

### 2.6. Statistical Analysis

The test results were expressed as the mean ± standard deviation (SD), calculated using Microsoft Excel (Microsoft, Redmond, WA, USA) software. The statistical significance was determined using the statistical analyses that were performed using Origin 2016 (OriginLab, Northampton, MA, USA). Here, *p* < 0.05 (*) indicates statistically significant, *p* < 0.01 (**) indicates very significant, and *p* < 0.001 (***) indicates extremely significant values.

## 3. Results and Discussion

The prepared foams of PU, 10HA/PU, and 20HA/PU have an interconnected porous structure shown by the micro-CT images in Figure 1A–F. The porosity decreases with the increase of the n-HA content. In Table 1, all of the measured porosities of the PU, 10HA/PU, and 20HA/PU scaffolds calculated using micro-CT are higher than that calculated by the liquid displacement method (LDM). The data difference between the micro-CT and LDM should be caused by the closed pores, in that the latter does not calculate the closed pores. Whether calculated by micro-CT or LDM, the porosity of the foams decreased with the increasing of the n-HA content, and the density value was the opposite (Table 1). The pore size ranged from 100 to 500 μm, as shown in the SEM photo in Figure 1G, which is suitable for new bone formation [29]. SEM–EDX gives the element mapping of calcium (Figure 1H) and phosphorus (Figure 1I), which indicates a uniform distribution of n-HA particles in the scaffold matrix. The TEM micrograph (Figure 1J) and the HRTEM image (Figure 1K) of the 10HA/PU composite further exhibit the presence of the rod-like n-HA particles, at a nanoscale of width 10–20 nm and length 50–100 nm, dispersed in the PU matrix homogenously, and no interface gap or phase separation is present. The nano-domain of the PU soft segments (in black) and hard segments (in white) were also displayed by the HRTEM image (Figure 1L), after the PU matrix was stained using ruthenium tetroxide. The high interpenetration and bridging of the soft and hard segments ensures the good quality and property of the PU matrix.

The XRD patterns, FTIR spectra, and XPS spectra were further tested to reveal the phase composition of the foams, and the chemical groups or chemical bonding of the n-HA and PU matrix. As we know, amorphous substances exhibit diffuses and diffraction patterns, whereas crystalline materials show strong diffraction peaks in the XRD spectrogram. In Figure 2A, the pure PU matrix shows only an envelope spectrum with a broad peak centered at 18°, whereas the n-HA particles exhibit a crystallized pattern with characteristic peaks at approximately 25.8°, 32°, 34°, 35.5°, and 40°, and so on. The XRD patterns of 10 HA/PU and 20 HA/PU still exhibit their individual peaks, except for a relative decrease in the peak intensity. The results make it clear that the composite is composed of HA and PU, and their initial composition and chemical structure are maintained in the composite foams [30,31,32].

The FTIR spectrum can be used to confirm the chemical groups or linkages within a polymeric structure, along with the extent of hydrogen bonding, conformation, and accessibility and interaction between the hard and soft segments in polyurethane [33]. The FTIR spectra obtained from the attenuated total reflection (ATR) methods in Figure 2B confirms the formation of urethane linkages by observing the respective peak positions. The peak at 3570 cm^−1^ represents the –OH groups of HA, and the peak around 3250 cm^−1^ represents the –NH groups of PU. The combined band at 3200–3500 cm^−1^ in the 10 HA/PU and 20 HA/PU composites is because of the overlapping of the –NH and –OH stretching vibrations. The band at 1070–1098 cm^−1^ is the overlapping of the PO_4_^3−^ groups of HA and the –O–C=O stretching vibrations of the PU polymer, and the band at 1020–1040 cm^−1^ also belongs to the PO_4_^3−^ groups. The peaks at 2945–2900 cm^−1^ and 2830–2800 cm^−1^ are ascribed to eh symmetric and asymmetric vibration of the CH_2_ groups of PU [34], which mostly represent the hard and soft segments, respectively. The bands at 1630–1660 cm^−1^ and 1680–1749 cm^−1^ represent the carbonyl C=O stretching of the allophanate or urethane (CONH) groups, and the bands at 1530–1550 cm^−1^ are from the −CN stretching/−NH bending [21]. The presence of the C=O, −CN/−NH, and –O–C=O characteristic bands could confirm the formation of urethane linkages in the n-HA/PU composite scaffolds. The weakening of the –OH peak at 3570 cm^−1^ and the free carbonyl peak at 1747 cm^−1^ in the composites indicates the formation of more hydrogen bonding [35], which could lead to strong interface and intermolecular interactions.

The XPS spectra of PU, 10 HA/PU, and 20 HA/PU are shown in Figure 2C–F. The chemical shifts of the binding energies of the pure PU and n-HA/PU demonstrated that the molecules of n-HA may have chemically bonded with the PU segment groups, as shown in Figure 2D–F. The fitted C1s peaks show that the peak intensity (or concentration) of biuret (CO–N–CO) is obviously increased after the addition of n-HA particles, and the binding energy increases from 283.5 eV of PU to 286.2 eV of 10 HA/PU and 286.3 eV of 20 HA/PU. It suggests that hydrogen bonds have been formed between the n-HA filler and PU matrix. The main peak of HN–CO–O and the small peak of CO–N–CO, in Figure 2D, at 282.1 eV and 283.5 eV, respectively, originate most likely from the allophanate or biuret, created by the reaction of the residual IPDI component with water (foaming agent) or urethane groups in PU. It is worth noting that the binding energy of the polar HN–CO–O and CO–N–CO groups of 10 HA/PU and 20 HA/PU shifts a lot compared with the pure PU, indicating that the addition of n-HA does affect the binding state or linkage of the PU polymer chains.

The dynamic mechanical analysis (DMA) is a useful technique for the evaluation of the viscoelastic property of polymers, by which a sinusoidal force (stress σ) is applied to the polymer, and the resulting displacement (strain) is measured [36]. When polymers composed of long molecular chains have unique viscoelastic properties, the characteristics of the elastic solids and Newtonian fluids combine. The temperature dependency of the storage modulus (E’), loss modulus (E”), and loss tangent (tan δ, from E”/E’) of the PU, 10 HA/PU, and 20 HA/PU are shown in Figure 3A–C, respectively. The storage modulus and loss modulus data are originated from the sample nature (elastic PU and rigid n-HA nanocrystals). The storage modulus and the loss modulus of the three foams drop sharply in the temperature range from −50 to 20 °C, in Figure 3A,B. The tan δ curves and data (<1) in Figure 3C indicate that the three foams are viscoelastic solid materials in the temperature range from 0 to 120 °C. The 20 HA/PU holds the highest storage modulus and loss modulus, and the lowest loss tangent, representing that it has a better viscoelasticity and is more rigid than pure PU and 10 HA/PU foams. The glass transition temperature (T_g_) values, in Figure 3C, of the 10 HA/PU and 20 HA/PU foams (~90 °C) are higher than that of the pure PU foam (~70 °C), suggesting an enhanced cross-linking in the HA/PU composites. Moreover, the 20 HA/PU sample shows the highest storage modulus resulting from its ability to resist intermolecular slippage, meaning more steric hindrance because of the stronger intermolecular interactions between the n-HA fillers and PU polymer chain. The broad tan δ peaks of 10 HA/PU and 20 HA/PU are indicative of a wider range of polymer chain branching within the PU matrix. The three PU foams also give a wide rubbery plateau, which is dependent on the molecular entanglements or cross-links, indicating that they have a good viscoelasticity at an applicable temperature [37,38].

Figure 3D shows the mechanical properties of PU, 10 HA/PU, and 20 HA/PU foams. It shows that the pure PU foam has a better elastomeric behavior because of its higher value of strain at break compared with the HA/PU foams; the strain at break for PU is approximately 1.5 times higher than that of 10 HA/PU, and 2.25 times higher than that of 20 HA/PU. The good elastomeric behavior of the PU foam results from the polyol as soft segments and the chain extender [39]. However, with the addition of n-HA particles, the tensile strength and modulus are apparently increased. The tensile strength of PU, 10 HA/PU, and 20 HA/PU is 2.97 MPa, 3.26 MPa, and 3.54 MPa, respectively, and the tensile modulus of the 10HA/PU and 20HA/PU foams are approximately 2.2 times and 3.4 times higher than that of the pure PU foam (*p* < 0.05). The reason for the increase in tensile strength and tensile modulus should be caused by the homodisperse or improved stress dispersity of the n-HA crystals in the PU matrix, and the tight interface bonding between them. It also results from the increased hydrogen bonds between the OH groups of the n-HA and the NCO groups of the IPDI in the n-HA containing polyurethanes [19]. Thus, the tight bonding interface becomes a dominating factor in the mechanical improvement of the HA/PU composites, as shown in the HRTEM image in Figure 1K, as well as in the analysis from XPS spectra in Figure 2. The additional amount of n-HA and the interfacial interaction between n-HA and PU are the major factors that determine the ultimate properties of the composite foams.

The differential scanning calorimetry (DSC) of the thermal analysis is a vital analytical method for understanding the structure–property relationships of different polymeric materials, and an effective method to evaluate materials’ thermal stability [40]. The DSC curves of pure PU and HA/PU composites are shown in Figure 3E. The thermal behavior for the three foams can be described in three endothermic and one exothermic process, as follows: I—the endothermic peak at 350 °C, in this process, the chemical bonds between the hard segment and soft segment break down; II—the endothermic peak around 400 °C indicates the breaking process of the hard segment, and the C–O bond of the carbamate groups in the main chain of PU matrix is destroyed, making the PU dissociate first into isocyanates and polyols; III—the endothermic peak from 420 to 430 °C represents the process that the long chain of soft segments break down and the isocyanates break down into carbon dioxide gas evolution; IV—the exothermic peak from 450 to 460 °C, during this period, the residual polyols break down into small molecules and release heat. All of the endothermic peaks and the exothermic peak are affected to some extent by the filling of the n-HA crystals when compared to the DSC curve of pure PU; in particular, the exothermic IV peak is largely strengthened for the two HA/PU composites, as a result of the increased hydrogen bonding between the –OH groups of HA and the HN–CO–O groups of the PU matrix, which enhanced the thermal stability of the composites.

The TG curve in Figure 3F presents a minor mass loss of approximately 4% over the temperature range of 50 to 200 °C, because of the loss of adsorbed water or carbon dioxide. The following thermal degradation of all of the samples is a two-step process, corresponding to the decomposition of the hard segments and soft segments in the PU matrix. In the first stage of decomposition (270–350 °C), the C–O bond of the carbamate groups in the main chain of the PU matrix is destroyed, making the PU dissociate first into isocyanates and polyols, then break down into a carbon dioxide gas evolution. The second stage of mass loss involves the further decomposition of the residual polyol of the PU matrix during 350–450 °C, which results in the 10 HA/PU composite retains nearly 10% of its weight, and the 20 HA/PU composite retains nearly 20% of its weight. The composition of n-HA is stable in such a temperature range. The result indicates that the addition of n-HA has a positive effect on the thermal stability of the PU matrix.

The LSCM images show the cell morphology of the BMSCs after four days of culture with the extraction solution of the three samples (A—blank control; B—PU; C—10HA/PU; D—20HA/PU). The cell morphology, including the red F-actin, blue nucleus, and green cytoplasm, can be clearly observed, as shown in Figure 4A–D. The normal cell spindle shape and well cell spreading with longer filopodia demonstrate that the composition of the n-HA and PU has no negative effect on the growth of the BMSCs.

The proliferation of BMSCs cultured on the PU, 10HA/PU, and 20HA/PU samples was determined using a CCK-8 assay. As shown in Figure 4E, the cell proliferation of all of the experimental groups steadily increases with culture time from day 1 to day 11, similar to the normal growth trend of the control. The cells cultured on 20 HA/PU show a significant increase compared with the other groups from day 1 to day 7. On day 11, the 10 HA/PU sample holds the highest proliferation, and the 20 HA/PU sample is still higher than the control. This indicates that both the 10 HA/PU and the 20 HA/PU foams display a good cytocompatibility, and can be used for the in vivo investigation.

The ALP activity and OCN content were also measured for the osteoblastic differentiation of the BMSCs cultured on different foams. Both the ALP activity (Figure 4F) and the OCN content (Figure 4G) indicate the successful osteoblastic transformation and good osteogenic activity of the BMSCs on the PU foams during the culture time, for up to 21 days. However, there are differences between the four groups, in which, the HA/PU foams exhibit much higher ALP and OCN activities than the pure PU foam and the control, and the 20 HA/PU foam shows the highest ALP and OCN expression. The ALP value of the 20 HA/PU group is approximately 1.5 times higher than that of the 10 HA/PU group and two times higher than that of pure PU and the control groups on day 21 *(p <* 0.001). The OCN value of the 20 HA/PU group is also much higher than that of the 10 HA/PU group, and the pure PU and control groups after cultured for seven days *(p <* 0.001). On day 21, the OCN value of the 20 HA/PU group is about two times higher than that of the 10 HA/PU group, and three times higher than that of the pure PU and the control groups *(p <* 0.001). The results indicate that the terminal marker OCN of osteoblastic differentiation can be up-regulated notably in the HA/PU composites compared with the pure PU, revealing that the presence of n-HA crystals in the matrix does promote the osteoblastic differentiation of BMSCs. Furthermore, each group was also observed with Trichrome dyeing after incubation for four days.

To compare the osteogenic capacity of the different foams, the PU and 20HA/PU foams were implanted into the mandible defects of New Zealand rabbits for 4 and 12 weeks. The histological sections of the harvested samples are shown in Figure 5. All of the three groups show an increasing growth of new bone during this time. However, the 20 HA/PU sample shows a better material–bone bonding interface. After four weeks of implantation, no inflammatory response could be observed in all of the groups, and the new bone tissue appeared in the defects of the control (Figure 5A), PU (Figure 5B), and 20HA/PU (Figure 5C). After four weeks, the new bone continuously grew with the implantation time (Figure 5D–F). Direct contact with the new bone can be observed for the 20 HA/PU sample (Figure 5F), while there is a gap present between the material and new bone for the PU sample (Figure 5D), suggesting that the 20 HA/PU sample has a better osteogenic capability because of the addition of n-HA crystals. The results of the physical and chemical analyses, in vitro cell culture, and the in vivo animal experiment indicate that the prepared HA/PU composite has a good flexibility, cytocompatibility, and bone-bonding bioactivity, which may be suitable for future plastic surgery and tissue regeneration. 

## 4. Conclusions

We designed and fabricated a porous system based on elastomer polyurethane, for potential use in plastic surgery. The n-HA/PU porous composite has good elasticity, flexibility, and supporting mechanical strength. The uniform distribution of the n-HA crystals in the PU matrix, while a homogeneous structure of hydrophilic and hydrophobic phase, was caused by hard and soft segments of PU. Furthermore, the tightly bonding interface on the nano particles and polymeric matrix is a dominating factor in the mechanical improvement of the HA/PU composites. The pore size (100–500 μm) and porosity (70%) are efficient for cell growth and for the regeneration of bone tissues, and the addition of n-HA fillers promoted cell infiltration and tissue regeneration as designed. The composite elastomers show good resilience to hold against the general press and tensile stress, as well as having the desired clinical manipulation. Further specific molding, such as the three-dimensional printing technique based on the n-HA/PU elastomer, will be a good candidate for plastic reconstruction.

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
