# Peer review of "Elastomeric Polyurethane Foams Incorporated with Nanosized Hydroxyapatite Fillers for Plastic Reconstruction"

_nanomaterials, 2018, doi:10.3390/nano8120972_

Reviewer 1 Report

Article "Elastomeric polyurethane foams incorporated with nanosized hydroxyapatite fillers for plastic reconstruction" is written in good langage and style, the results are presented in logic sequece and clearly. Article consiste from 2 seperate parts: 1) PU chemistry and technology (or material science) and 2) experiments with obtained material "in vitro", "in vivo". Sorry, I`m only expert in the 1st part. I hope, that other reviewer (-s) the 2nd part will review more carrefully, than me.

Comments, questions and suggestions.

L.44 never before HA with full names

L.48 PLGA and PCL, not explained with full names

Check the whole text about abbreviations! For example L.54 and 59 in both: PU!

L.74-78 As catalyst you used DBTL, it is tin containing! Ithink it isn`t the better material which potentially will be used in medicine (surgery)!

Why the maximum content of HA in PU is 20%. Why not higher? Maybe some problems with viscosity, but it isn`t described!

L.94 RuO4 - subscrip

L.100-101 … cm-1   - superscrip

Check whole text. Maybe some convertation problems, but in few places instead of subscript or superscript is flat line!

L.107 What is Jupiter? Netzsch is Netzsch, but why Jupiter?

L.158 and 166. The same equipment twice in long brackets!

L.217 What is 10~20 nm: between figures symbol approx.!

Tab.1 Less significant digits in values and errors (standart deviations). One significant digit in error (standart deviatin) and the same precisity in value! 78 +/- 5; 71 +/- 2; 62 +/- 4.

Fig.1 Few figures (E; H; I) are practcally Black rectangle!

L.234 ...spectrogram31. I think it is some mistake. The same in L.310.

L.268. ...about 282.1 and 283.5.   About is 282 and 284. 282.1 and 283.5 is accurate!

Fig.3 In caption better: … the tensile properties (D) …

In Fig.3, D on Y-axis Tensile strength (now is strenth, without g)

L.290 What is Tt, never before it isn`t explained.

L.315-323 and Fig.3E. There are no any valuable info from DSC measurements. In your explanations: 1 peak at 350, the 2nd at 400, … No any explanations, what are the molecular processes influenced peaks. I recomend exclude the DSC experiments from article.

Conclusions.

There are no any sentence about these In vitro in vivo experiments!

Author Response

Response to Reviewer 1 Comments

Point 1: L.44 never before HA with full names. L.48 PLGA and PCL, not explained with full names. Check the whole text about abbreviations! For example L.54 and 59 in both: PU!

Answer: Sorry about these mistakes. The full names of HA, PLGA and PCL are added in L.44 and L.48 as the following. And the PU has been changed according to the advice.

Excellent biocompatibility and bone-bonding bioactivity of synthetic hydroxyapatite (HA) made it attractive in plastic surgery.

In recent years, hybrid composites composed of polymers with nanosized hydroxyapatite (n-HA), may create a bio-mimetic interface to host tissue and can be considered as one of the most promising biomaterials. Different n-HA/polymer composites such as n-HA added Poly(lactic-co-glycolic acid) (PLGA) or Polycaprolactone (PCL) have been developed and broadly investigated as scaffolds for bone repair [11-13].

It is well known that there is a class of polymer materials which can be synthesized as “soft or hard scaffolds” by different segmented structure [16]. The synthetic polymers called polyurethane (PU) have been applied in tissue engineering with the advent of a wide range of mechanical and physical properties by changing the ratio of soft and hard segments and composition [17,18]. PU exhibit diverse properties with self-foaming porous and unique segmented structure, excellent and varying mechanical properties, and good biocompatibility, which could be considerable interest to specific tissue [19,20].

Point 2: L.74-78 As catalyst you used DBTL, it is tin containing! I think it isn`t the better material which potentially will be used in medicine (surgery)!

Answer: Thanks for the reviewer’s suggestion. In this study, we used DBTL in a small dose as the catalyst, and the cell experiments in vitro and animal test in vivo showed the synthetic scaffolds did not show apparent toxicity. Considering the inherent toxicity of organotin, we will try to find other catalyst as a safe alternative with low toxicity.

Point 3: Why the maximum content of HA in PU is 20%. Why not higher? Maybe some problems with viscosity, but it isn`t described!

Answer: It’s exactly as review’s said, higher content of inorganic HA will cause higher viscosity that effects the elasticity of the elastomer. According to the advice, the description has been added in the paper as the following:

Considering higher content of inorganic HA will produce stiff scaffold which is not proper to the plastic elastomer, the maximum content of HA has been set to 20 wt%.

Point 4: L.94 RuO4 – subscript; L.100-101 … cm-1 - superscript. Check whole text. Maybe some convertation problems, but in few places instead of subscript or superscript is flat line!

Answer: Sorry for these mistakes. I have checked the whole text and marked the correction as the following.

The microphase image of the PU scaffolds was labeled by a staining agent of ruthenium tetroxide (RuO4) which the staining and exposure time was 15 min [25].

The chemical groups of porous materials were recorded by Fourier transform infrared spectroscopy (FTIR, Nicolet 6700, Nicolet Perkin Elmer Co., USA) at a wavenumber range of 650 cm−1-4000 cm−1.

An intramuscular injection of 1 × 105 units penicillin was given to each rabbit during initial 3 days after implant operation. At the time points of 4 and 12 weeks, the rabbits were sacrificed via CO2 asphyxiation and the samples with surrounding tissue were harvested.

Point 5: L.107 What is Jupiter? Netzsch is Netzsch, but why Jupiter?

Answer: The model of thermal analyzer instrument is STA 449 F3 Jupiter. Thanks for the advice, I have added the full information about the instrument as the following:

The thermal stability of PU composites was measured by Thermogravimetric analysis (TGA) simultaneous thermal analyzer (STA 449 F3 Jupiter, NETZSCH, Germany) under nitrogen flow rate at 20 ml/min.

Point 6: L.158 and 166. The same equipment twice in long brackets!

Answer: Thanks for your kind reminder. The second one is deleted

The morphology and spreading of the BMSCs growing in the extracted solution of samples was imaging using a laser scanning confocal microscopy (LSCM, Nikon, A1R, MP+, Japan). The BMSCs cultured with the extracted solution at 37°C for 96 h in a humidified atmosphere of 5% CO2 and 95% air, in between, changed extracted solution in two days. After incubation, the cells were fixed in 3.7% (w/v) formaldehyde solution, permeabilized with 0.2% (v/v) Triton X-100/phosphate buffered saline (PBS), and stained successively with 0.25 μM MitoTracker® Green FM probes for chondriosome, 0.15 μM Alexa Fluor® 532 phalloidin for red F-actin, and 5 μg/mL blue fluorescent Hoechst 33342 in 1% (w/v) bovine serum albumin (BSA)/PBS for nucleus, according to the manufacturer’s instructions of Thermo Fisher Scientific Co.. The cells were then washed with PBS to remove excess stains. The labeled cells were imaged using LSCM.

Point 7: L.217 What is 10~20 nm: between figures symbol approx.!

Answer: Sorry for the mistake. I have revised it as follows.

The TEM micrograph (J) and the HRTEM image (K) of 10HA/PU composite further exhibit the presence of the rod-like n-HA particles at nanoscale of width 10 - 20 nm and length 50 - 100 nm dispersed in the PU matrix homogenously, and no interface gap or phase separation is present.

Point 8: Tab.1 Less significant digits in values and errors (standard deviations). One significant digit in error (standard deviation) and the same precision in value! 78 +/- 5; 71 +/- 2; 62 +/- 4.

Answer: Thanks for the suggestion. I have revised the data as your advice.

Sample

Porosity by Micro-CT (%)

Porosity by LDM   (%)

Density by Micro-CT g/cm3

Density by LDM

g/cm3

PU

80.57

78± 5

0.16

0.17± 0.02

10HA/PU

75.26

71± 2

0.24

0.24± 0.02

20HA/PU

65.95

62± 4

0.34

0.35± 0.01

Point 9: Fig.1 Few figures (E; H; I) are practically Black rectangle!

Answer: Sorry for these obscure figures. I have adjusted the contrast ratio.

Point 10: L.234 ...spectrogram31. I think it is some mistake. The same in L.310. L.268. ...about 282.1 and 283.5. About is 282 and 284. 282.1 and 283.5 is accurate!

Answer: Thanks for your reminder. I have checked and revised according to your advice.

The fitted C1s peaks show that the peak intensity (or concentration) of biuret (CO-N-CO) is obviously increased after addition of n-HA particles and the binding energy increases from 283.5 eV of PU to 286.2 eV of 10HA/PU and 286.3 eV of 20HA/PU. The main peak of HN-CO-O and the small peak of CO-N-CO in Figure 2D at 282.1 eV and 283.5 eV respectively originate most likely from the allophanate or biuret, created by the reaction of residual IPDI component with water (foaming agent) or urethane groups in PU.

Point 11: Fig.3 In caption better: … the tensile properties (D) … In Fig.3, D on Y-axis Tensile strength (now is strenth, without g)

Answer: Sorry for the mistake. I have changed the “mechanical properties” to “tensile properties” and “strenth” to “strength”.

Figure 3. Dynamic mechanical analysis of storage modulus (A), loss modulus (B) and loss tangent (C) against temperature, and the tensile properties (D), DSC curves (E) and TG curves (F) of PU, 10HA/PU and 20HA/PU. Significantly different at *p<0.05, **p<0.01, **p<0.001.< span="">

Point 12: L.290 What is Tt, never before it isn`t explained.

Answer: Sorry for the typing mistake. I have changed “Tt” to “Tg”.

The glass transition temperature (Tg) values in Figure 3C of 10HA/PU and 20HA/PU foams (~90 °C) are higher than that of pure PU foam (~70 °C), suggesting an enhanced cross-linking in the HA/PU composites.

Point 13: L.315-323 and Fig.3E. There is no any valuable info from DSC measurements. In your explanations: 1 peak at 350, the 2nd at 400, … No any explanations, what are the molecular processes influenced peaks. I recomend exclude the DSC experiments from article. Conclusions. There are no any sentence about these In vitro in vivo experiments!

Answer: Thanks for your suggestion. We need describe more clearly about DSC curve because it presents some important information of materials. We have added some explanations for the peaks below.

The thermal behavior for the three foams can be described in three endothermic and one exothermic process: I – the endothermic peak at 350 °C, in this process, the chemical bonds between hard segment and soft segment broke down; II – the endothermic peak around 400 °C indicates the breaking process of hard segment; the C-O bond of the carbamate groups in the main chain of the PU matrix is destroyed, making the PU dissociate first into isocyanates and polyols; III – the endothermic peak from 420 °C to 430 °C represents the process that long chain of soft segment broke down and isocyanates broke down into carbon dioxide gas evolution.; and IV – the exothermic peak from 450 °C to 460 °C, during this period, residual polyols broke down into small molecules and released heat. All the endothermic peaks and the exothermic peak are affected to some extent by the filling of n-HA crystals when compared to the DSC curve of pure PU, especially, the exothermic IV peak is largely strengthened for the two HA/PU composites, resulting from the more hydrogen bonding between the -OH groups of HA and HN-CO-O groups of PU matrix which enhanced the thermal stability of the composites.

For conclusion, there has a sentence about these In vitro in vivo experiments (blue color). We added some sentence to emphasis the good bioactivity results of HA added polyurethane (yellow color).

The pore size (100-500 μm) and porosity (70%) are efficient for cells growth and regeneration of bone tissues, and the addition of n-HA fillers promoted cell infiltration and tissue regeneration as designed.

Reviewer 2 Report

This manuscript presents the development of an elastomeric polyurethane composite containing nanosized hydroxyapatite. The authors present a thorough materials characterization along with in vitro and in vivo characterization of the osteogenic capabilities of rat bone marrow MSCs. The work is interesting and addressing my comments will strengthen the manuscript.

Comments

1. The results presented in Fig. 4 are presented and described out of sequence. The authors highlight the performance of the BMSCs in osteogenic differentiation in Fig. 4B, 4C before presenting the basic cytotoxicity trial of exposing the BMSCs to leachate from the samples in Fig. 4D-F. The authors also do not clearly describe the 4 day treatment in the results and discussion. Based on the text, a reader may assume that the cell morphologies shown resulted from culture on the scaffolds. The authors should clarify the sequence and presentation of these results.

2. The overlapping XRD patterns in Fig. 2A is hard to read. I recommend to space the patterns out to allow the reader to clearly observe the patterns.

3. Please add a legend or label for Fig. 2B. I assume the labels are the same as Fig. 2A, but it should be presented in the same panel.

4. The labels in Fig. 3D are not legible. A larger font should be used. Also, “Broken Elongation” should be changed to elongation at failure or something similar.

5. How many rabbits were used per group and time point? Were the sample materials paired in rabbits (i.e. two control materials implanted in animal)?

Minor Comments

1. Passage not generation for BMSCs.

2. The ALP and OCN assays should be normalized to total protein content from the samples using a BCA assay or similar analysis.

3. There were a few references cited that were not formatted correctly in the text, i.e. just the number presented without [ ].

Author Response

Response to Reviewer 2 Comments

Point 1: The results presented in Fig. 4 are presented and described out of sequence. The authors highlight the performance of the BMSCs in osteogenic differentiation in Fig. 4B, 4C before presenting the basic cytotoxicity trial of exposing the BMSCs to leachate from the samples in Fig. 4D-F. The authors also do not clearly describe the 4 day treatment in the results and discussion. Based on the text, a reader may assume that the cell morphologies shown resulted from culture on the scaffolds. The authors should clarify the sequence and presentation of these results.

Answer: Sorry for the mess. We should arrange the content more logically and have revised it according to your advice as below.

Figure 4. The CLSM images showing the cell morphology of BMSCs after 4 days of culture with extraction solution of the three samples (A- blank control; B-PU; C-10HA/PU; D-20HA/PU) F-actin stained in red, mitochondria stained in green, and the nuclei stained in blue. The cell proliferation of BMSCs cultured on PU, 10HA/PU and 20HA/PU samples via the CCK-8 assay (E), and the expression of alkaline phosphatase (F) and osteocalcin (G) of BMSCs cultured with the three samples for 7, 14 and 21 days. (*p<0.05, **p<0.01, ***p<0.001).< span="">

The CLSM images showing the cell morphology of BMSCs after 4 days of culture with extraction solution of the three samples (A- blank control; B-PU; C-10HA/PU; D-20HA/PU). The cell morphology including the red F-actin, blue nucleus and green cytoplasm can be clearly observed, as shown in Figure 4 (A-D).The normal cell spindle shape and well cell spreading with longer filopodia demonstrate that the composition of the n-HA and PU has no negative effect on the growth of BMSCs.

The proliferation of BMSCs cultured on PU, 10HA/PU and 20HA/PU samples was determined by CCK-8 assay. As shown in Figure 4E, the cell proliferation of all experimental groups steady increases with culture time from day 1 to day 11, similar to the normal growth trend of the control. The cells cultured on 20HA/PU shows a significant increase compared to other groups from day 1 to day 7. On day 11, the 10HA/PU sample holds the highest proliferation, and the 20HA/PU sample is still higher than the control. This indicates that both the 10HA/PU and the 20HA/PU foams display good cytocompatibility, and can be used for in vivo investigation.

The ALP activity and OCN content were also measured for osteoblastic differentiation of BMSCs cultured on different foams. Both the ALP activity (4F) and the OCN content (4G) indicate successful osteoblastic transformation and good osteogenic activity of the BMSCs on PU foams during the culture time up to 21 days. However, there are differences between the four groups, in which, the HA/PU foams exhibit much higher ALP and OCN activities than pure PU foam and the control, and the 20HA/PU foam shows the highest ALP and OCN expression. The ALP value of 20HA/PU group is approximately 1.5 times higher than that of 10HA/PU group and 2 times higher than that of pure PU and the control groups on day 21 (p<0.001). The OCN value of 20HA/PU group is also much higher than that of 10HA/PU group, and pure PU and control groups after cultured for 7 days (p<0.001). On day 21, the OCN value of 20HA/PU group is about 2 times higher than that of 10HA/PU group and 3 times higher than that of pure PU and the control groups (p<0.001). The results indicate that the terminal marker OCN of osteoblastic differentiation can be up-regulated notably in the HA/PU composites compared to pure PU, revealing that the presence of n-HA crystals in PU matrix does promote the osteoblastic differentiation of BMSCs. Furthermore, each group was also observed with Trichrome dyeing after incubation for 4 days.

Point 2: The overlapping XRD patterns in Fig. 2A is hard to read. I recommend to space the patterns out to allow the reader to clearly observe the patterns.

Answer: Thanks for your suggestion. We have spaced the patterns out, hope it could be clearer for read.

Figure 2. The XRD patterns (A), FTIR spectra (B), and XPS spectra of C1s peak (C) and the peak fitting of PU (D), 10HA/PU (E) and 20HA/PU (F).

Point 3: Please add a legend or label for Fig. 2B. I assume the labels are the same as Fig. 2A, but it should be presented in the same panel.

Answer: Thanks for your kind reminder. I have named each curve, according to your advice. You can see figure 2.

Point 4: The labels in Fig. 3D are not legible. A larger font should be used. Also, “Broken Elongation” should be changed to elongation at failure or something similar.

Answer: Sorry for the mistake. I have changed the “Broken Elongation” to “Elongation at break”and make the labels legible and enlarged the fonts in Figure 3.

Point 5: How many rabbits were used per group and time point? Were the sample materials paired in rabbits (i.e. two control materials implanted in animal)?

Answer: Sorry for the missing information. We have added the relative information as the following:

Twelve rabbits have used in the test. The implantation periods were set for 4 and 12 weeks. At each time point, two rabbits were in each group and three groups have been operated according to control, PU and 20HA/PU groups.

Minor Comments

1. Passage not generation for BMSCs.

Answer: Thanks for the advice. We have change “generation” to “passage”

Bone marrow mesenchymal stem cells (BMSCs) isolated from Sprague Dawley rats (1-month-old, male) with a weight of approximately 100 g were provided by West China Animal Center of Sichuan University, and the third passage of BMSCs were utilized in the experiments [26, 27].

2. The ALP and OCN assays should be normalized to total protein content from the samples using a BCA assay or similar analysis.

Answer: It is a good question for us. We did not think it over and only tested the ALP and OCN according to previous study (Li, L. M.; Zuo, Y.; Zou, Q.; Yang, B. Y.; Lin, L. L.; Li, J. D.; Li, Y. B., Hierarchical Structure and Mechanical Improvement of an n-HA/GCO-PU Composite Scaffold for Bone Regeneration. Acs Appl Mater Inter 2015, 7 (40), 22618-22629). We used the ALP and OCN Kit to quantify the total concentration of the two proteins at the different time point. In this study, the results can show the relative trend of cells differentiation on foams. And the method can also explain that the foams have different influence on cell differentiation. In further study, we will be careful on the question and do it better according to your advice.

Cell differentiation: Quantitative analysis of ALP and OCN activity for BMSCs cells cultured on the three scaffolds by using Enzyme-linked Immuno Sorbent Assay (ELISA) has been employed to assess the Osteogenic differentiation study [21].

3. There were a few references cited that were not formatted correctly in the text, i.e. just the number presented without [ ].

Answer: Sorry for the missing information. I have checked and revised it.

It also results from the increased hydrogen bonds between the OH groups of n-HA and the NCO groups of IPDI in the n-HA containing polyurethanes [19].
